# OpenReview forum: "Feature Augmentation of GNNs for ILPs: Local Uniqueness Suffices"
_ICLR.cc/2026/Conference — Submitted to ICLR 2026_

### Official Review · Reviewer_myYa · 2025-10-22

**Soundness:** 2
**Presentation:** 2
**Contribution:** 3
**Rating:** 2
**Confidence:** 4

**Summary:**

This paper proposed a coloring method as feature augmentation for learning to optimize ILP problems. First, the graph is properly colored so that every two d-neighbors of the same node acquire different colors, then the color information is encoded and treated as extra node features. The GNN then performs message passing with the augmented features. The authors show expressivity results and generalization bound.

**Strengths:**

- The motivation is novel and clear, instead of coloring the nodes differently in the same orbits, the authors proposed to color the nodes differently in d-neighbors, which is more parsimonious and requires fewer colors, and intuitively can generalize better.
- The proposed method is very simple and neat, should be easy to implement.
- The proposed method looks promising on some of the experiments.

**Weaknesses:**

- I believe the paper writing can be improved a lot, some proofs can be moved the appendix, while some concepts should be better explained in the main paper. The paper seems to be written in haste, and a lot of concepts are not well explained, which limits readers' understanding of the paper. Also see my questions below.
- The greedy coloring algorithm seems expensive when the d grows, as the degree of the power graph will be very high.
- The experiment part also seems rushed out in haste. The authors reported results on d=1, but I believe they should test more on the d hyperparameter, as it is the most crucial parameter of the whole paper. And the experiments should repeat with a couple of random seeds.
- __The derivation of the generalization bound is totally missing, no appendix attached!__

**Questions:**

- Is observation 1 supported by clear theory? Or just because some prior work all used shallow GNNs? If the latter, it is quite convincing to stick to shallow GNNs.
- The authors proposed a greedy method to color the graph. Is the greedy approach characterized in terms of optimality? If the greedily colored graph uses too many unnecessary colors, the generalization bound might be violated.
- The authors stated in the intro that: "ColorGNN is valueinvariant: it automatically embeds colors without requiring predefined numeric encodings..." But I cannot find how ColorGNN is actually implemented. The emb function in line 193 is unclear to me.
- The authors claim that ColorUID is a light-weight version of ColorGNN, what is exactly the difference? In my opinion this should be clearly stated in the paper, in a mathematically way instead of just a few sentences.
- It is not quite motivated to me why authors tested generalization performance on LPs, as far as I know LPs do not require such colorings to break the symmetry. And it makes little sense to test on Zinc. To test the generalization performance, it makes more sense to train on one ILP dataset and test on another.

---

> ### Author Response · Authors · 2025-11-25
> **Official Response to Reviewer myYa part 1**
>
> We are grateful to the reviewer for raising this issue. Your comment has helped us improve the presentation of our contributions. We have thoroughly revised the paper following your comments: **all formal proofs have been moved to the appendix**. We also provide a more detailed derivation of the generalization bound in the appendix(see Appendix B.4). In addition, we added an ablation study on the effect of the coloring radius $d$ to better support our claims.
>
> >Q1:The greedy coloring algorithm seems expensive when the d grows, as the degree of the power graph will be very high.
>
> A1:We agree that the cost of coloring grows with d.
> 1)From a theoretical standpoint, the degree of the $2d$-power graph is at most $\Delta_{2d}\leq|V|$, **so the complexity of our greedy scheme is $O(|V|\cdot\Delta_{2d})\leq O(|V|^2)$ in the worst case**.
> 2)In practice, our method does **not** require running a full $d$-hop unique coloring for a $d$-layer GNN. As shown in our BPP ablation (varying the coloring radius), a 2-layer GNN achieves the best performance with 4-hop coloring (corresponding to 2-hop unique coloring), but 2-hop coloring(corresponding to 1-hop unique coloring) already offers a **good trade-off between accuracy and computational cost**.
> |  d | Top-30%  | Top-50%   |  Top-70%   | Top-90%   | Top-100%   |
> |---|---|---|---|---|---|
> |  1 |  6.05 |   10.03|  14.92 | 18.88  |  28.87 |
> |  2 |  0.00 |  0.01 |  1.02 |  7.71 |  18.27 |
> |  3 |   0|  0.01 |  0.83 |  7.81 |  18.26 |
> |  4 |   0|   0|  **0.68** | 5.39  |  **16.34** |
> |  5 |  0 |  0 |  0.72 |  **5.31** |  16.67 |
> |  6 |  0 |  0.01 |  0.70 |  5.43 | 16.41 |
>
> This indicates that we can often work with a **smaller effective radius**, for which $\Delta_{2d}$ is much smaller than $|V|$.
> 3)We also report the **actual runtime** and **average number of nodes** on our ILP datasets:
> |   | BPP  | BIP  |SMSP|
> |---|---|---|---|
> |  # of Nodes |460| 1278  |45854.1|
> | time(s)  | 0.003  | 0.028  |2.952|
>
> On the largest graphs from SMSP (around 45k nodes), the average coloring time is about 2.95 seconds. Since coloring is performed once as an **offline preprocessing** step, and **not recomputed during each training iteration**, this overhead is acceptable in our setting.
>
> >Q2:The experiment part also seems rushed out in haste. The authors reported results on d=1, but I believe they should test more on the d hyperparameter, as it is the most crucial parameter of the whole paper. And the experiments should repeat with a couple of random seeds.
>
> A2:Thank you for the question. In the revised version, **we have added an ablation study that varies the coloring radius d to better support our method.** On the BPP dataset, we observe that: (1) **4-hop coloring yields (nearly) the best performance**, which is consistent with our theory that a 2-layer GNN only requires information up to 4 hops; (2) 2-hop coloring is not optimal in terms of accuracy, but it provides a **reasonable trade-off between computational cost and performance**; and (3) as the coloring radius increases **beyond 4, performance starts to deteriorate**. Based on these observations, our practical **guideline** is to choose a coloring radius that is **not larger than twice the GNN depth**. The detailed results on BPP are reported as follows：
> |  d | Top-30%  | Top-50%   |  Top-70%   | Top-90%   | Top-100%   |
> |---|---|---|---|---|---|
> |  1 |  6.05 |   10.03|  14.92 | 18.88  |  28.87 |
> |  2 |  0.00 |  0.01 |  1.02 |  7.71 |  18.27 |
> |  3 |   0|  0.01 |  0.83 |  7.81 |  18.26 |
> |  4 |   0|   0|  **0.68** | 5.39  |  **16.34** |
> |  5 |  0 |  0 |  0.72 |  **5.31** |  16.67 |
> |  6 |  0 |  0.01 |  0.70 |  5.43 | 16.41 |
>
> Regarding randomness, all our ILP/LP experiments closely **follow the original experimental protocol and codebase of prior work**, which report single-run results with fixed seeds. **To ensure strict comparability, we kept this setting for the main tables**. In addition, we now report a 5-seed study on BPP (each entry averaged over 5 different seeds), which shows that the performance trends with respect to $d$ are **stable**. Subsequently, we will gradually extend the multi-seed evaluations to all datasets.

---

> ### Author Response · Authors · 2025-11-25
> **Official Response to Reviewer myYa part 2**
>
> >Q3:The derivation of the generalization bound is totally missing, no appendix attached!
>
> A3:Thanks for you comments. We provide a **more detailed derivation of the generalization bound in the appendix(see Appendix B.4)**.
>
> > Q4:Is observation 1 supported by clear theory? Or just because some prior work all used shallow GNNs? If the latter, it is quite convincing to stick to shallow GNNs.
>
> A4:It is the **latter**. Observation 1 is not introduced as a new theoretical result, but as an empirical design choice aligned with existing practice in L2O. Recent works on L2O for ILP and related problems [1,2,3,4,5,6,7] all use shallow GNNs, typically with 2 message-passing layers.
> [1]Exact combinatorial optimization with graph convolutional neural networks
>
> [2]A gnn-guided predict-and-search framework for mixed-integer linear programming
>
> [3]Contrastive predict-and-search for mixed integer linear programs
>
> [4]Pdhg-unrolled learning-to-optimize method for large-scale linear programming
>
> [5]When gnns meet symmetry in ilps: an orbit-based feature augmentation approach
>
> [6]Apollo-milp: An alternating prediction-correction neural solving framework for mixed-integer linear programming
>
> [7]Symilo: A symmetry-aware learning framework for integer linear optimization
>
>
> >Q5:The authors proposed a greedy method to color the graph. Is the greedy approach characterized in terms of optimality? If the greedily colored graph uses too many unnecessary colors, the generalization bound might be violated.
>
> A5:
> Q:Thank you for the question. Our greedy procedure is characterized in a standard way: it achieves the classical $\Delta+1$ upper bound on **the number of colors, which is worst-case optimal for polynomial-time algorithms (even though it is not instance-optimal)**. Computing the **minimum possible number of colors for a given graph is NP-hard**, so our **theory uses the number of colors $|{C}|$ as a guidance** term in the generalization bound, while in practice we rely on the greedy heuristic.
> Empirically, the greedy algorithm **does not blow up the number of colors**. In our ablation on the BIP dataset, the largest graphs have 1278 nodes; the greedy 1-hop coloring uses 114 colors, while the optimal 1-hop unique coloring cannot use fewer than 106 colors (because there exists a constraint node connected to 106 variables). So at most 8 colors are redundant in this case. For BPP and SMSP, we still observe a substantial reduction in the number of UIDs compared to global-UID baselines, and the **performance gains from this reduction appear to dominate any minor effect from a small amount of redundancy in the greedy coloring**.The detailed results regarding the number of UIDs on the ILP datasets are reported as follows:
> | Method   |  BPP |   BIP |   SMSP   |
> |----------|-----:|------:|---------:|
> | Uniform  |  420 | 1082.41 | 23137.25 |
> | Position |  420 | 1083    | 23417.97 |
> | Orbit    |  140 | **51**  | 1000     |
> | Orbit+   |  140 | **51**  | 6070     |
> | ColorGNN | **32** | 114  | **171.68** |

---

> ### Author Response · Authors · 2025-11-25
> **Official Response to Reviewer myYa part 3**
>
> >Q6:The authors stated in the intro that: "ColorGNN is valueinvariant: it automatically embeds colors without requiring predefined numeric encodings…" But I cannot find how ColorGNN is actually implemented. The emb function in line 193 is unclear to me.
>
> A6:
> Q： In ColorGNN we i**mplement value invariance by grouping embeddings by color**. In the vanilla case without colors, all nodes share a single learnable linear embedding function. In ColorGNN, each node v has a color $C(v)$, and we maintain $|{C}|$separate embedding modules $\{\mathrm{emb}_{C_{(v)}}\}$. **Node $v$ is embedded by applying the embedding corresponding to its color, i.e.,$\mathrm{emb}_{C(v)}$**. Equivalently, nodes are partitioned into $|{C}|$ groups by color; nodes **within the same group share parameters**, while different color groups use different embeddings. For more details, please refer to our open-source code: https://anonymous.4open.science/r/dhop_Coloring4GNN-B062/
>
> A7: Thank you for the suggestion. In the revised version, we now give a precise mathematical description of both **ColorGNN** and **ColorUID**, and clarify in what sense ColorUID is a light-weight variant.
> We start from a standard message-passing GNN. Let $L_v$ denote the input features of node $v$, and let $N_1(v)$ be the 1-hop neighborhood of $v$. A $d$-layer GNN is defined by:
> 1. Initialization
>    $$
>    h_v^{(0)} := \text{emb}(L_v),
>    $$
> 2. Message passing
>    for $k = 1,\dots,d$,
>    $$
>    h_v^{(k)} := \text{Merge}^{(k)}\Bigl(
>        h_v^{(k-1)},\;
>        \text{Aggr}\bigl(\{\,h_u^{(k-1)},\,W(u,v)\,\} \mid u \in N_1(v)\bigr)
>    \Bigr),
>    $$
> 3. Output each node $v$ outputs $h_v^{(d)}$.
>
> In the paper we consider three ColorGNN variants (plus one additional variant (ColorGNN(AGGR)) introduced during the rebuttal), all of which are parallel architectural choices that differ only in where the color information enters the network. Let $C(v)$ denote the color of node $v$.
>
> **ColorGNN**
>
>   We modify the initialization step as
>   $$
>   h\_v^{(0)} := \text{emb}\_{C(v)}(L_v),
>   $$
> where each $\text{emb}_{C(v)}$ is a learnable linear transformation specific to color $C(v)$. Nodes with the same color share parameters, and different colors use different embedding functions.
>
> **ColorGNN(Merge)**
>
>   We instead color the merge function in the message-passing step:
>   $$h\_v^{(k)} := \text{Merge}^{(k)}\_{C(v)}\Bigl(
>       h\_v^{(k-1)},\;
>       \text{Aggr}\bigl(\{\,h\_u^{(k-1)},\,W(u,v)\,\} \mid u \in N\_1(v)\bigr)
>   \Bigr),
>   $$
>   where $\text{Merge}^{(k)}_{C(v)}$ is a learnable mapping specific to color $C(v)$.
>
> **ColorGNN(AGGR)**
>
>   Here we color the aggregation operator itself:
>   $$
>   h\_v^{(k)} := \text{Merge}^{(k)}\Bigl(
>       h\_v^{(k-1)},\;
>       \text{Aggr}\bigl(\{\,\alpha\_{C(u)} \cdot h\_u^{(k-1)},\,W(u,v)\,\} \mid u \in N\_1(v)\bigr)
>   \Bigr),
>   $$
>   where $\alpha_{C(u)} \in \mathbb{R}$ is a learnable scalar specific to color $C(u)$. We initialize $\alpha_{C(u)} \sim \mathcal{N}(1,1)$ for all $u$.
>
> **ColorUID**
>
>   In ColorUID, colors are injected purely as *additional input features*. Concretely, we modify the initialization step as
> $$
>   h\_v^{(0)} := \text{emb}\bigl(L_v,\, C(v)\bigr).
>   $$

---

> ### Author Response · Authors · 2025-11-25
> **Official Response to Reviewer myYa part 4**
>
> A8:Thank you for the question.
> 1) After observing substantial gains on ILP tasks, **we turned to their natural relaxations, LPs**. It is indeed known that LP learning does not require additional IDs to break symmetry [1]. However, in practice LP instances **vary dramatically in size**, and scaling GNN-based solvers to much larger instances is a central challenge [2,3]. Our LP experiments therefore focus on an OOD setting where the model is trained on small LPs and tested on significantly larger ones. In this regime, **we find that adding local UIDs markedly improves OOD generalization, which we believe provides useful insight for scaling-oriented LP research, even if symmetry breaking is not strictly necessary.**
> 2) **More generally**, we also want to verify that our local UID scheme is **compatible with standard, non-ILP GNN tasks**. For this reason, we include experiments on ZINC, a widely used regression benchmark for graph-level prediction. On ZINC we simply plug our local UIDs into an existing SOTA architecture (PEARL) and observe consistent improvements. **The goal here is not to claim any special connection between ZINC and ILPs**, but to **demonstrate that our design principle** extends beyond ILP/LP to conventional GNN benchmarks.
> 3) Regarding the suggestion to “train on one ILP dataset and test on another”: in our view, ILP and LP instances from different problem families have extremely different structures and distributions, so training on problem A and testing on problem B is an extremely challenging transfer setting. We believe a more natural first step is to study generalization across scales within the same problem family (small to large instances), where the distribution shift is still substantial but more controlled.  Our ILP and LP experiments are therefore explicitly designed around this intra-family scaling scenario.
>
> [1]On representing linear programs by graph neural networks
>
> [2]Pdhg-unrolled learning-to-optimize method for large-scale linear programming
>
> [3]Practical large-scale linear programming using primal-dual hybrid gradient

---

> > ### Comment · Reviewer_myYa · 2025-11-25
> >
> > Thanks for the detailed reply and the refined submission. The ablation experiments looks cool. Most of my answered have been answered, yet the proof is in the appendix is still unreadable. I understand it is impossible to have a perfect version in such short time. I would like to slightly raise my score, and looking forward to a complete version by camera ready!

---

### Official Review · Reviewer_qakW · 2025-10-24

**Soundness:** 4
**Presentation:** 3
**Contribution:** 3
**Rating:** 6
**Confidence:** 3

**Summary:**

This paper introduces a novel feature augmentation scheme for Graph Neural Networks (GNNs) applied to Integer Linear Programming (ILP) problems. The authors propose a "Local-UID" (Unique Identifier) scheme based on d-hop unique coloring, where nodes only need unique identifiers within their d-hop neighborhoods rather than globally. They introduce two architectures: ColorGNN (which uses color-conditioned embeddings) and ColorUID (a lightweight variant). The key insight is that for d-layer GNNs, local uniqueness within d-hop neighborhoods provides the same expressive power as global UIDs while requiring fewer identifiers and offering better generalization.

**Strengths:**

* The paper provides rigorous theoretical analysis showing that Local-UIDs achieve the same expressive power as Global-UIDs (Theorem 1) while offering better generalization bounds
* The greedy d-hop coloring algorithm (Algorithm 1) is efficient and practical, with clear complexity bounds.
* The evaluation covers multiple ILP benchmarks and demonstrates strong out-of-distribution generalization on LP tasks.
* The approach significantly reduces the number of required identifiers (e.g., 32 vs 420 on BPP dataset), which aligns well with the principle of parsimony in feature augmentation.
* ColorGNN is value-invariant and can be seamlessly combined with other feature augmentation techniques like Orbit and Orbit+.

**Weaknesses:**

* On SMSP dataset, the method underperforms Orbit/Orbit+ on Top-30% and Top-50% metrics, suggesting limitations when instance-specific information is crucial.
* The paper mainly focuses on one specific way of incorporating colors (through embeddings). More architectural variants could strengthen the contribution.

**Questions:**

* How does performance change with different values of d? Is there an optimal d for different problem types?
* How sensitive is the method to different color assignments? Would different runs of the greedy algorithm produce significantly different results?
* What is the actual runtime overhead of computing d-hop colorings on the largest graphs in your experiments?
* Beyond ILPs and LPs, what other optimization problems could benefit from this approach?

---

> ### Author Response · Authors · 2025-11-25
> **Official Response to Reviewer qakW part 1**
>
> We sincerely thank the reviewer for the thoughtful . The comments have helped us a lot. We address each concern in detail below.
> >Q1:On SMSP dataset, the method underperforms Orbit/Orbit+ on Top-30% and Top-50% metrics, suggesting limitations when instance-specific information is crucial.
>
> A1: Thank you for the observation. 1) Orbit/Orbit+ explicitly exploit **instance-level information** via **variable symmetry detection**, while our method starts from anonymous graphs and only uses structural information without any instance-specific symmetry preprocessing; 2) We view combining our local UID scheme with such instance-specific information as a very promising future direction.
>
> >Q2: The paper mainly focuses on one specific way of incorporating colors (through embeddings). More architectural variants could strengthen the contribution.
>
> A2:
> Thanks for your insightful comments.
> 1)Our goal is to propose a general design guideline for incorporating colors. From an expressivity perspective, injecting colors at the input layer is already sufficient while keeping the number of parameters small.
> 1)Following your suggestion, we have added a new architectural variant, **ColorGNN(AGGR)**, which applies coloring in the aggregation function. Concretely:
> -  ${h}^{(0)}_v:=\text{emb}({L}_v)$,
> - For $k=1$ to $d$, compute
>  $${h}_{v}^{(k)}:=\text{Merge}^{(k)}\left({h}_{v}^{(k-1)},\text{Aggr}(\{\alpha_{C(u)}\cdot{h}^{(k-1)}_{u},W(u,v)\}\mid u\in N_1(v)\right),
> $$
> where $\alpha_{C(v)}\in R$ is a learnable scalar specific to color $C(v)$, initialized as $\alpha_{C_{(u)}} \sim \mathcal N{(1,1)}$  for all $u$
> - Finally, each node $v$ outputs ${h}^{(d)}_v$.
> Empirically, ColorGNN(AGGR) **performs comparably to our other variants** while using fewer parameters. It is also **compatible with other augmentation methods**(Orbit/Orbit+), although its performance is slightly weaker, possibly because the color-dependent aggregation partially breaks the symmetry of the aggregation operator.
> The detailed results on BPP are reported as follows：
>
> |  method | Top-30%  | Top-50%   |  Top-70%   | Top-90%   | Top-100%   |
> |---|---|---|---|---|---|
> |Non-Aug  | 5.81 |  9.74 |  14.43 |  18.28 |  27.94 |
> |ColorUID  | 0.00 |  0.01 |  **1.02** |  7.71 |  18.27 |
> |   ColorGNN|   0.00|  0.01 | 0.98 |  7.80 | 18.29 |
> |   ColorGNN(Merge)|   0.00|  0.02 |  1.04 | 7.69 |18.28 |
> |   ColorGNN(Aggr)|   0.00|  0.035 |  1.08 |  11.10 |  20.91 |
> |   ColorGNN(Aggr),Orbit|   0.01|  0.035 |  1.80 |  7.19 |  **18.19** |
> |   ColorGNN(Aggr),Orbit+|   0.00|  0.01 |  1.23 | **5.82** |  19.41 |
>
>
> >Q3:Questions:How does performance change with different values of d? Is there an optimal d for different problem types?
>
> A3:Thank you for the question. In the revised version, we have **added an ablation study** that varies the coloring radius d to better support our method. On the BPP dataset, we observe that: (1) **4-hop coloring yields (nearly) the best performance**, which is consistent with our theory that a 2-layer GNN only requires information up to 4 hops; (2) 2-hop coloring is not optimal in terms of accuracy, but it provides a **reasonable trade-off** between computational cost and performance; and (3) as the coloring radius increases beyond 4, performance** starts to deteriorate**. Based on these observations, our practical guideline is to choose a coloring radius that is not larger than twice the GNN depth. **The optimal choice of d for different problem types likely varies and would require further empirical investigation.**
> The detailed results on BPP are reported as follows：
> |  d | Top-30%  | Top-50%   |  Top-70%   | Top-90%   | Top-100%   |
> |---|---|---|---|---|---|
> |  1 |  6.05 |   10.03|  14.92 | 18.88  |  28.87 |
> |  2 |  0.00 |  0.01 |  1.02 |  7.71 |  18.27 |
> |  3 |   0|  0.01 |  0.83 |  7.81 |  18.26 |
> |  4 |   0|   0|  **0.68** | 5.39  |  **16.34** |
> |  5 |  0 |  0 |  0.72 |  **5.31** |  16.67 |
> |  6 |  0 |  0.01 |  0.70 |  5.43 | 16.41 |

---

> ### Author Response · Authors · 2025-11-25
> **Official Response to Reviewer qakW part 2**
>
> A4: Thank you for the question. Due to time constraints, we **conducted a robustness study only on the BPP dataset**, where all reported numbers are averaged over 5 different random seeds. The results show that both **ColorUID and ColorGNN exhibit small variance** across runs, indicating that they are **not very sensitive** to the specific color assignments produced by the greedy algorithm. In contrast, **Orbit/Orbit+ display noticeably larger variance**, and methods that combine our approach with Orbit/Orbit+ inherit this higher variability. The detailed results on BPP are reported as follows：
>  |  method | Top-30%  | Top-50%   |  Top-70%   | Top-90%   | Top-100%   |
> |---|---|---|---|---|---|
> Non-Aug | 5.989±0.120 | 9.973±0.156 | 14.775±0.230 | 18.728±0.299 | 28.633±0.462
> Orbit | 0.001±0.002 | 0.015±0.014 | 2.061±0.324 | 7.874±0.348 | 17.615±0.327
> Orbit+ | 0.000±0.000 | 0.001±0.002 | 1.277±0.100 | 7.091±0.183 | 19.237±0.127
> |ColorUID | 0± 0 |0.011 ± 0.004| 0.936 ± 0.106 |7.792 ± 0.061 |  18.260 ± 0.039 |
> ColorGNN | 0±0 | 0.008±0.006 | 0.920±0.100 | 7.821±0.049 | 18.351±0.085|
> ColorGNN,Orbit | 0.000±0.000 | 0.026±0.011 | 1.165±0.102 | 6.640±0.141 | 17.425±0.110
> ColorGNN,Orbit+ | 0.000±0.000 | 0.000±0.000 | 1.061±0.145 | 6.169±0.221 | **16.856±0.224**
>
>
> >Q5:What is the actual runtime overhead of computing d-hop colorings on the largest graphs in your experiments?
>
> A5: The largest graphs in our experiments come from the SMSP dataset. We report, for all three ILP datasets, the **average number of nodes** as well as the **average actual runtime** required to compute the corresponding d-hop colorings. The detailed results on BPP are reported as follows：
> |   | BPP  | BIP  |SMSP|
> |---|---|---|---|
> |  # of Nodes |460| 1278  |45854.1|
> | time(s)  | 0.0026  | 0.028  |2.952|
>
> Note that all colorings are performed only during the data preprocessing stage; during training, there is no need to recompute the coloring at each iteration.
>
> >Q6:Beyond ILPs and LPs, what other optimization problems could benefit from this approach?
>
> A6: Thank you for the question; we agree this is an important direction for future work. Our approach **only relies on the underlying graph structure**, so in principle **any optimization problem that can be modeled as a graph could benefit from ColorGNN/ColorUID** as an expressivity-enhancing component. In particular, learning-based methods for **Quadratic Programs** (QPs) and Quadratically **Constrained Quadratic Programs** (QCQPs) [1,2] could directly incorporate our local UID scheme. Moreover, ILPs already appear in a wide range of applications such as scheduling, allocation, and planning, so the **proposed method is applicable to many practically relevant optimization tasks.**
>
> [1]Expressive Power of Graph Neural Networks for (Mixed-Integer) Quadratic Programs
>
> [2]On Representing Convex Quadratically Constrained Quadratic Programs via Graph Neural Networks

---

### Official Review · Reviewer_rUR6 · 2025-11-02

**Soundness:** 2
**Presentation:** 1
**Contribution:** 1
**Rating:** 2
**Confidence:** 3

**Summary:**

This paper seeks to address the challenge of solving integer linear programming (ILP) problems using Graph Neural Networks (GNNs). Since conventional GNNs are limited in their expressivity, the authors note that augmenting nodes with unique identifiers is useful but not generalizable. To address this limitation, they propose a Local-UID scheme based on d-hop uniqueness coloring, which ensures that identifiers are unique only within each node’s d-hop neighborhood. The paper provides some theoretical justification along with experimental analysis to validate the method.

**Strengths:**

1. The motivation of restricting node IDs to local d-hops is interesting and, to my knowledge, has not been studied previously.
1. The writing of the paper is fairly clear, although the presentation could be improved.

**Weaknesses:**

1. The paper is not clearly positioned. The abstract/introduction mentions the task of ILP as the main motivating factor. However, as the method develops, it appears the authors are addressing general GNN challenges. It is unclear why ILPs are taken as the main task rather than conventional tasks that have clear benchmarks for evaluation.
1. If ILPs are the main motivation, why do the authors test on the ZINC dataset, which has no relevance to the method?
1. The evaluation of the method is not comprehensive, with very few baselines included. The authors need to position the paper more clearly and thoroughly analyze the method across a variety of settings to demonstrate its benefits.
1. The theoretical development of the idea is somewhat straightforward. I do not see any specific results that are non-trivial and reveal deeper insights.

**Questions:**

Please see weaknesses.

---

> ### Author Response · Authors · 2025-11-25
> **Official Response to Reviewer rUR6**
>
> We thank the reviewer for their thoughtful comments and suggestions.
> >Q1:The paper is not clearly positioned. The abstract/introduction mentions the task of ILP as the main motivating factor. However, as the method develops, it appears the authors are addressing general GNN challenges. It is unclear why ILPs are taken as the main task rather than conventional tasks that have clear benchmarks for evaluation.
>
> >Q2:If ILPs are the main motivation, why do the authors test on the ZINC dataset, which has no relevance to the method?
>
> A1&A2: Our primary **motivation is indeed ILP**: on ILP graphs without IDs, the expressive power of standard GNNs is severely limited.**Our goal is to understand and improve this specific setting**, and the design of our local UID scheme is driven by the structure of ILP / bipartite graphs and the needs of learning-to-optimize pipelines.
> At the same time, the mechanism we propose (parsimonious local UIDs via d-hop colorings) **is not tied to ILPs and directly addresses a more general question** about how to inject IDs/positional information into GNNs. This is why, after establishing results on ILP benchmarks, we deliberately step out of the ILP domain:
> - **LP experiments** show that the same scheme yields strong in- and out-of-distribution generalization on 2 LP problems.
> - **ZINC experiments** serve as a sanity check that our method is compatible with standard GNN benchmarks and architectures: we **simply plug** our local UIDs into an existing SOTA model on ZINC (rather than designing a new molecule-specific architecture), and observe consistent gains. **This is not because ZINC is “about ILPs”**, but to demonstrate that the proposed local UID principle **is broadly applicable and competitive** on conventional tasks as well.
>
> >Q3:The evaluation of the method is not comprehensive, with very few baselines included. The authors need to position the paper more clearly and thoroughly analyze the method across a variety of settings to demonstrate its benefits.
>
> A3:We appreciate this concern. In the ILP setting, we follow the standard experimental protocol used in prior work and include, to the best of our knowledge, the most widely adopted baselines in this area: **Orbit, Orbit+, Position, and Uniform**. These are evaluated across three types of problems and six datasets (BPP, BIP, SMSP, WA, PageRank, and ZINC). For the “general GNN” setting, we further adopt **PEARL** as the SOTA baseline on ZINC and simply plug our local UIDs into it. We additionally include **weak coloring** [2] as another local-ID baseline, and compare it directly with our method on BPP; the detailed BPP results are reported as follows：
> |  method | Top-30%  | Top-50%   |  Top-70%   | Top-90%   | Top-100%   |
> |---|---|---|---|---|---|
> Non-Aug | 5.989±0.120 | 9.973±0.156 | 14.775±0.230 | 18.728±0.299 | 28.633±0.462
> WeakColor | 6.066±0.013 | 10.061±0.008 | 14.890±0.011 | 18.878±0.005 | 28.873±0.002
> |ColorUID | 0± 0 |0.011 ± 0.004| 0.936 ± 0.106 |**7.792 ± 0.061** |  **18.260 ± 0.039** |
> ColorGNN | 0±0 | **0.008±0.006** | **0.920±0.100** | 7.821±0.049 | 18.351±0.085|
>
> During the rebuttal period, beyond adding new baseline experiments, we also incorporated: a stability analysis of our method, runtime measurements for coloring the largest graphs, an ablation study of performance with respect to the parameter d, and comparison experiments with a new variant, ColorGNN (AGGR).
>
> >Q4:The theoretical development of the idea is somewhat straightforward. I do not see any specific results that are non-trivial and reveal deeper insights.
>
> A4: Thank you for the comment. Our primary goal in this work is not to develop a purely theoretical framework, but **to propose a practical method** that improves the expressivity and generalization of GNNs on ILP-style tasks. The theoretical results are included to **justify and support the design**, rather than to serve as the main contribution of the paper.

---

### Official Review · Reviewer_jhyf · 2025-11-03

**Soundness:** 3
**Presentation:** 3
**Contribution:** 2
**Rating:** 4
**Confidence:** 3

**Summary:**

It is known that node IDs can help the expressivity of GNNs but the paper argues that global IDs may harm generalization, hence it propsoes the use of local identifiers. Specifically, it proposes a unique local coloring of nodes for up to d hops. That means that each node in a d-hop neighborhood has a unique color. This is done via a greedy algorithm and yields substantial savings compared to global IDs.  This is then used to inform architecture design for a specific GNN. It is shown that a d layer GNN with local (d hop) ids is just as expressive as one with global ids.
The paper proves expressivity and generalization results and experiments on solving IPs and LPs.`

**Strengths:**

- The paper is clearly written and the different elements of the paper clearly support its main position.
- The paper provides a good mix of theoretical and empirical support for its method.
- The empirical performance looks solid and the theoretical results (I didn't check proofs thoroughly) make sense.

**Weaknesses:**

- The motivation is a little vague. It is claimed that global IDs can lead to spurious correlations but is there any study in the literature that corroborates that? It seems you could argue about the merits of local features purely from an efficiency standpoint. But the paper actually proposes a design principle. I think for this kind of proclamation the paper requires stronger empirical evidence or more references to related work that support this position. For the record, I do think it's a reasonable proposal but if it's to be proposed as a design principle it should be well supported.
- Discussion of related work seems inadequate. Previous papers have talked about GNN expressivity in various contexts (including solving combiantorial problems) so some comments on how this paper differs or connects to existing approaches would go a long way, e.g., [1] uses also local features and proves their expressivity. Also more discussion on relationships to other works like [2,3])  would be good. How does the local coloring proposed here compare to the weak coloring proposed in the paper by Sato et al. If those ideas are related I think it is important to connect to this literature even though those papers are not specifically about solving IPs. Similarly, what about positional encodings for GNNs as discussed in [4] (and references therein)? On that note, having a standard strong baseline such as Signnet[5] in the experiments would be nice.
- For a d-layer GNN, do you need d-hop colorings or can you go way lower than that in practice? This perhaps connects to my comments about weak coloring above. This is a central contribution of the paper so I would be curious to see an ablation that compares to other local colorings.
- In table 3 WA OOD MSE seems to vary quite a bit. How is it possible that it goes from 10^2 to 10^-1? Is the model training at all when there are no local IDs?

Overall, this is an interesting paper that I am open to accepting. I start with a tentative score that I will update after the rebuttal.

1. Vignac, Clement, Andreas Loukas, and Pascal Frossard. "Building powerful and equivariant graph neural networks with structural message-passing." Advances in neural information processing systems 33 (2020): 14143-14155.
2. Sato, Ryoma, Makoto Yamada, and Hisashi Kashima. "Approximation ratios of graph neural networks for combinatorial problems." Advances in Neural Information Processing Systems 32 (2019).
3. Sato, Ryoma, Makoto Yamada, and Hisashi Kashima. "Random features strengthen graph neural networks." Proceedings of the 2021 SIAM international conference on data mining (SDM). Society for Industrial and Applied Mathematics, 2021.
4. Grötschla, Florian, Jiaqing Xie, and Roger Wattenhofer. "Benchmarking positional encodings for gnns and graph transformers." arXiv preprint arXiv:2411.12732 (2024).
5. Lim, Derek, et al. "Sign and basis invariant networks for spectral graph representation learning." arXiv preprint arXiv:2202.13013 (2022).

**Questions:**

see above

---

> ### Author Response · Authors · 2025-11-25
> **Official Response to Reviewer jhyf part 1**
>
> Thank you for your thoughtful comments and suggestions.
> >Q1:The motivation is a little vague. It is claimed that global IDs can lead to spurious correlations but is there any study in the literature that corroborates that? It seems you could argue about the merits of local features purely from an efficiency standpoint. But the paper actually proposes a design principle. I think for this kind of proclamation the paper requires stronger empirical evidence or more references to related work that support this position. For the record, I do think it's a reasonable proposal but if it's to be proposed as a design principle it should be well supported.
>
> A1:
> We agree that such a claim should be well supported, and we have revised the paper accordingly.
> First, we have **expanded the discussion in the Introduction to better connect our motivation to existing work on UIDs and spurious correlations**. Several recent papers explicitly document that global node IDs dramatically increase the risk of memorization and unstable behavior: (i) [1] shows that once UIDs are available, a GNN can fit essentially arbitrary labelings on the training distribution. This immediately implies that **if the ID-generation mechanism changes even slightly at the test time, the learned model can fail catastrophically**. (ii) [2] demonstrates that different ID assignments of the same underlying graph can lead to different graph-level predictions, **indicating that the model is relying on ID-specific artifacts rather than the underlying structure**. (iii) [3] provides **empirical evidence** that GNNs equipped with global UIDs tend **to overfit**, which harms their generalization performance.
> Second, beyond the efficiency benefits of local features, our experiments directly compare global UID schemes (Uniform, Position, Orbit/Orbit+) with our local-UID constructions (ColorGNN / ColorUID) on both ILP/LP tasks and an additional molecular benchmark. In-distribution, global UIDs can sometimes match or slightly improve performance, but on out-of-distribution LP instances and larger graphs, their performance degrades sharply, whereas our local-UID variants remain stable or even improve.
>
> [1]On the universality of graph neural networks on large random graphs
>
> [2]Towards invariance to node identifiers in graph neural networks
>
> [3]On the Utilization of Unique Node Identifiers in Graph Neural Networks
>
> >Q2:Discussion of related work seems inadequate. Previous papers have talked about GNN expressivity in various contexts (including solving combiantorial problems) so some comments on how this paper differs or connects to existing approaches would go a long way, e.g., [1] uses also local features and proves their expressivity. Also more discussion on relationships to other works like [2,3]) would be good. How does the local coloring proposed here compare to the weak coloring proposed in the paper by Sato et al. If those ideas are related I think it is important to connect to this literature even though those papers are not specifically about solving IPs. Similarly, what about positional encodings for GNNs as discussed in [4] (and references therein)? On that note, having a standard strong baseline such as Signnet[5] in the experiments would be nice.
>
> A2: We have **expanded the related work section** to discuss [1,2,3,4,5] and to clarify how our approach relates to these lines of work.
> On the experimental side:
> 1. We have **added a comparison with SignNet** [5] as a strong baseline (see the Table 6).
> 2. We implemented the weak coloring scheme from [2] and compared it with our method on the BPP dataset, which is a standard ILP benchmark. **Its performance is worse, and in fact even inferior to using no augmentation at all**. Our interpretation is that ILP instances are naturally modeled as bipartite graphs (variable nodes vs. constraint nodes). For a bipartite graph,**weak coloring assigns a single color to each side**, causing all variable nodes to share the same color and thus **remain indistinguishable**, even though predicting the solution crucially requires distinguishing individual variables. The detailed results are presented as follows:
> 3.
>  |  method | Top-30%  | Top-50%   |  Top-70%   | Top-90%   | Top-100%   |
> |---|---|---|---|---|---|
> Non-Aug | 5.989±0.120 | 9.973±0.156 | 14.775±0.230 | 18.728±0.299 | 28.633±0.462
> WeakColor | 6.066±0.013 | 10.061±0.008 | 14.890±0.011 | 18.878±0.005 | 28.873±0.002
> |ColorUID | 0± 0 |0.011 ± 0.004| 0.936 ± 0.106 |**7.792 ± 0.061** |  **18.260 ± 0.039** |
> ColorGNN | 0±0 | **0.008±0.006** | **0.920±0.100** | 7.821±0.049 | 18.351±0.085|
>
> [1]Building powerful and equivariant graph neural networks with structural message-passing
>
> [2]Approximation ratios of graph neural networks for combinatorial problems
>
> [3]Random features strengthen graph neural networks
>
> [4]Benchmarking positional encodings for gnns and graph transformers
>
> [5]Sign and basis invariant networks for spectral graph representation learning

---

> ### Author Response · Authors · 2025-11-25
> **Official Response to Reviewer jhyf part 2**
>
> >Q3:For a d-layer GNN, do you need d-hop colorings or can you go way lower than that in practice?This perhaps connects to my comments about weak coloring above. This is a central contribution of the paper so I would be curious to see an ablation that compares to other local colorings.
>
> A3:
> Thank you for the insightful question.
> From a theoretical perspective, o**ur use of $d$-hop unique colorings (equivalently, a proper coloring on the $2d$-hop graph) is worst-case optimal** for a $d$-layer GNN: if we require all nodes within  hops to have distinct colors, then our greedy scheme achieves the optimal asymptotic bound on the number of colors in the worst case. For a  specific graph, however, fewer colors may be sufficient in practice, but finding the m**inimum such coloring is NP-hard**, so we rely on an efficient heuristic rather than trying to aggressively minimize the number of colors instance by instance.
> At the same time, we do not strictly need full $d$-hop uniqueness in all applications. In many ILP/LP settings, the input features already distinguish nodes sufficiently, or IDs are barely needed at all (e.g., some LP tasks). In such cases, weaker local colorings (with radius < $d$) can still work reasonably well in practice.
> To support our claims, we have added an ablation on the BPP dataset using ColorUID, where we vary the coloring radius for a fixed 2-layer GNN. The results are consistent with our theory: using $4$-hop coloring (i.e., 2-hop unique coloring) yields the best performance, but smaller radius slightly reduce accuracy while lowering computational cost. In our final setup, we therefore choose a 2-hop coloring (1-hop unique coloring) as a practical trade-off between performance and efficiency.The detailed results are presented as follows:
> |  d | Top-30%  | Top-50%   |  Top-70%   | Top-90%   | Top-100%   |
> |---|---|---|---|---|---|
> |  1 |  6.05 |   10.03|  14.92 | 18.88  |  28.87 |
> |  2 |  0.00 |  0.01 |  1.02 |  7.71 |  18.27 |
> |  3 |   0|  0.01 |  0.83 |  7.81 |  18.26 |
> |  4 |   0|   0|  **0.68** | 5.39  |  **16.34** |
> |  5 |  0 |  0 |  0.72 |  **5.31** |  16.67 |
> |  6 |  0 |  0.01 |  0.70 |  5.43 | 16.41 |
>
> >Q4:In table 3 WA OOD MSE seems to vary quite a bit. How is it possible that it goes from 10^2 to 10^-1? Is the model training at all when there are no local IDs?
>
> A4:
> Thank you for pointing this out. All WA experiments strictly follow **the original training setup** of [1], using the same hyperparameters, and in all configurations the models are **trained to convergence**. The large change in OOD MSE is due to the evaluation protocol: the GNN is trained on small-size WA instances and tested on substantially larger ones. For these LP problems, **increasing the problem size drastically changes the scale and distribution of the optimal solutions**. An OOD MSE on the order of $10^2$ indicates that, **without local IDs, the GNN essentially fails** to extrapolate to larger instances and produces predictions that are off by orders of magnitude. In contrast, when local IDs are used, the model can still capture the solution distribution reasonably well, which yields OOD MSE around $10^{-1}$.
>
> [1]Pdhg-unrolled learning-to-optimize method for large-scale linear programming

---

### Meta-Review · Area_Chair_3vkK · 2026-01-08

**Summary:**

The paper proposes a feature augmentation for GNNs. In particular, it constructs d-hop locally unique node colors and injects them to improve expressivity and generalization in learning-to-optimize ILP settings. Reviewers provided detailed feedback and the authors made a clear effort in the rebuttal.

However, the rebuttal and revisions still read somewhat rushed, and several substantive concerns remain insufficiently resolved. In particular, the paper’s positioning is still unclear (general GNN design principle vs. L2O-specific contribution); The theoretical component also remains contentious: one reviewer still found the appendix proofs “unreadable.” Additionally, while coloring runtime is reported, it is still not clearly contextualized against GNN inference time and baseline runtimes. These issues may be addressable with more time and careful rewriting, but in its current form the paper does not yet feel ready for publication.

**Reviewer Concerns:**

### [jhyf]

All concerns are properly addressed.

### [rUR6]

Q1 & Q2: The authors' arguments regarding the motivation and the ZINC experiments make sense. However, their defense actually reinforces the need for a major revision of the writing. The current manuscript leaves an unclear impression regarding the scope of the contribution: is the proposed coloring method novel for the entire GNN community, or is it strictly a contribution to Learning to Optimize (L2O)?
* If the method is novel and effective for general graph tasks, significantly more experiments and evidence are needed to support this broader claim, and the narrative needs rewriting to frame integer optimization as just one application.
* If the method is novel for general GNNs but primarily works for L2O, a comprehensive analysis of the reasons is required, which also necessitates rewriting the narrative.
* If the method is only new to L2O but has been proposed or studied in general GNN literature, this distinction must be explicitly discussed in the introduction and contributions.

Q3: The authors' answer addressed the concern.

Q4: The authors' answer makes sense, but by acknowledging that the theoretical results are straightforward and serve merely as justification, this lowers the paper's overall contribution.

### [qakW]

Q1: The authors' response is reasonable, but they merely acknowledge the limitation rather than resolving or mitigating it.

Q5: While the coloring running time is reported, it remains unclear how this cost compares to the GNN inference time and the runtime of other baseline algorithms.

Other concerns are addressed.

### [myYa]

Most of the reviewer’s concerns were effectively addressed, but the theoretical proof moved to the appendix remained a point of contention and was explicitly described as “unreadable,” likely due to hasty preparation and a reliance on high-level citations rather than a self-contained argument. For instance, the expressivity argument (Theorem 1) glosses over the critical neighborhood reconstruction step by delegating it to external references, while Proposition 2 relies on compressed reasoning and ambiguous notation. Furthermore, the derived generalization bound is misaligned with the experiments: the theory explicitly models graph-level binary classification, whereas the actual ILP benchmarks evaluate node-level solution prediction.

**Reviewer Scores:**

Reviewer jhyf is likely to increase their score from 4 to 6, as their concerns are addressed and they expressed openness to reevaluation.

Reviewer rUR6 is expected to retain the score of 2, as the major concern remains unaddressed.

Reviewer qakW is likely to maintain the score of 6; although most concerns are addressed, the context suggests the score will hold.

Reviewer myYa is likely to increase from 2 to 4, having explicitly stated an intention to slightly raise the score after most concerns were resolved.

---

### Decision · Program_Chairs · 2026-01-26

Reject